# 30×30 biodiversity gains rely on national coordination

Isaac Eckert [1,2] ✉, Andrea Brown[1,2], Dominique Caron [1,2], Federico Riva[3] & Laura J. Pollock [1,2] ✉

Global commitments to protect 30% of land by 2030 present an opportunity to combat the biodiversity crisis, but reducing extinction risk will depend on where countries expand protection. Here, we explore a range of 30×30 conservation scenarios that vary what dimension of biodiversity is prioritized (taxonomic groups, species-at-risk, biodiversity facets) and how protection is coordinated (transnational, national, or regional approaches) to test which decisions influence our ability to capture biodiversity in spatial planning. Using Canada as a model nation, we evaluate how well each scenario captures biodiversity using scalable indicators while accounting for climate change, data bias, and uncertainty. We find that only 15% of all terrestrial vertebrates, plants, and butterflies (representing only 6.6% of species-at-risk) are adequately represented in existing protected land. However, a nationally coordinated approach to 30×30 could protect 65% of all species representing 40% of all species-at-risk. How protection is coordinated has the largest impact, with regional approaches protecting up to 38% fewer species and 65% fewer species-at-risk, while the choice of biodiversity incurs much smaller trade-offs. These results demonstrate the potential of 30×30 while highlighting the critical importance of biodiversity-informed national strategies.

Protected areas are pivotal to biodiversity conservation[1], but existing networks are inadequate when it comes to safeguarding Earth's biodiversity under climate change, lowering extinction rates, and preventing the erosion of critical ecosystem services[2–5]. Fortunately, over 190 nations have committed to protecting 30% of their land by 2030 (30 × 30), providing what could be humanity's last chance to prevent the catastrophic loss of global biodiversity by protecting land that serves biodiversity today and into the future[6,7]. But to facilitate positive outcomes for nature, we need to take stock of what is already protected, estimate what could be protected under 30 × 30, and understand how different conservation priorities and strategies influence our ability to capture biodiversity in protected areas[2].

Identifying land for protection is a long-standing challenge in conservation[8]. Historically, protected areas were often founded to protect landscapes, not biodiversity[4]. Where biodiversity has been considered, protected areas usually target specific taxa like migratory birds, species at-risk, or important lineages[9,10]. However, thanks to recent leaps in biodiversity science and data availability, spatial planning can now consider a broad range of biodiversity[11,12]. Biodiversity indicators can be mapped for thousands of species and functional and phylogenetic facets link taxa to functioning ecosystems, future option values, and millions of years of evolutionary history[13,14]. But with the proliferation of new ways of considering biodiversity, comes a critical need to determine the trade-offs associated with protecting some elements of biodiversity over others.

Similarly, how nations choose to coordinate the expansion of protected areas can influence spatial priorities and biodiversity outcomes[15,16]. Although full transnational coordination would optimally protect biodiversity in the broadest sense[14], coordination at the national scale, where federal governments organize protection within their borders to fulfill their commitment to 30 × 30, is far more feasible. But even within nations, conservation regularly happens at regional scales

[1]Dept. of Biology, McGill University, H3A 1B1 Montreal, QC, Canada. [2]Quebec Center for Biodiversity Science, Montreal, QC, Canada. [3]Institute for Environmental Studies, Vrije Universiteit Amsterdam, Amsterdam, The Netherlands. ✉e-mail: isaac.eckert@mail.mcgill.ca; laura.pollock@mcgill.ca

and spatial representation of protected areas (parochialism) across sub-national political or ecological jurisdictions is a substantial requirement for 30 × 30[7]. However, regional conservation initiatives can easily contrast with national or transnational priorities and surprisingly, the trade-offs associated with coordinating protection at different spatial scales remain largely unexplored, despite their central importance to the success of targets like 30 × 30 for biodiversity[17–19].

Here, we explore how different approaches to reaching 30 × 30 incur trade-offs and impact our ability to protect a broad range of biodiversity. Using Canada, a nation with considerable conservation potential[20,21] and a rapidly changing climate[22], we quantify how well existing protected areas capture biodiversity, how much could be protected under 30 × 30, and test whether prioritizing different elements of biodiversity versus coordinating protection at different spatial scales matters more for our ability to capture biodiversity in spatial planning. Using Zonation 5[23], we simulate 30 × 30 expansion scenarios that each protect 30% of land but vary what element of biodiversity is prioritized and how protection is coordinated spatially (Fig. 1). Since effective spatial planning must incorporate climate change[3], we prioritized "win-win" areas within species' current ranges that remain climatically viable into the future, thereby identifying climate refugia while also achieving species-level complementarity and irreplaceability[24]. To evaluate scenarios, we calculated trade-offs as the difference in potential protection between every scenario and the optimal national scenario which best protects Canadian biodiversity, both in terms of the number of species considered "protected" using a Species Protection Index (SPI)[25] and the amount of biodiversity captured (weighted endemism)[26]. We find that how protection is coordinated across spatial scales matters far more than what dimension of biodiversity is prioritized when it comes to capturing biodiversity in spatial planning and safeguarding species into the future under climate change. These findings confirm the critical need for national strategies for reaching 30×30 and demonstrate the importance of quantifying indicators and assessing trade-offs to facilitate biodiversity-informed conservation planning.

## Results

### Poor existing protection enables large conservation gains under 30 × 30

We find that existing protected areas, representing 15.4% of intact terrestrial land (excluding areas of high human footprint and ceded Indigenous land), do not effectively capture biodiversity and only protect (based on SPI) 15.1% of all terrestrial vertebrates, plants, and butterflies. Currently protected species represent only 6.6% of nationally listed species at-risk and only 1% of amphibians and reptiles, which amounts to just a single species (*Lithobates sylvaticus*). However, large conservation gains are possible if protected areas are expanded to reach 30%. Since 30 × 30 is a national target and biodiversity indicators are often reported at national scales to track international progress, the optimal scenario for Canada is the national scenario, which includes all terrestrial vertebrates (*n* = 697), plants (*n* = 3378), and butterflies (*n* = 190), prioritizes land at the national scale, and balances protection across all species. Reaching 30×30 under this nationally coordinated strategy could quadruple the number of protected species to protect 65% of all taxa (Fig. 2a). This increase in protection represents a 6-fold increase in the number of protected species at-risk (protecting 40%) and a 54-fold increase in the number of protected amphibians and reptiles (protecting 60%).

Relative to this optimal *National* scenario, alternative scenarios that prioritize different elements of biodiversity (taxonomic groups, species at-risk, or biodiversity facets) or that vary the scale at which protection is coordinated, incur variable trade-offs (Supplementary Data 1). Trade-offs were negligible for scenarios that prioritize *Amphibian & Reptile*, *Plant*, *Butterfly*, *Functional* and *Phylogenetic diversity*, and *National Species at-Risk*, meaning a similar number of species were protected compared to the optimal national scenario. Trade-offs were minimal for *Mammal*, and *Global Species-at-Risk* scenarios (1–5% fewer species protected relative to optimal) while prioritizing *Birds* or coordinating protection at *Transnational* scales incurs moderate trade-offs (10.9% and 16.3% fewer species protected respectively). Finally, scenarios that coordinate protection at regional scales, by conducting independent prioritizations for *Provinces & Territories* or *Ecozones* separately, incur the most severe trade-offs (33.2% and 37.5% fewer species protected). High priority areas for all scenarios are visualized in Fig. S1.

### How protection is coordinated matters far more that what biodiversity is prioritized

Overall, the scale at which protection is coordinated had the largest impact on the outcomes for biodiversity. Compared to the optimal *National* scenario, prioritizing land independently within Provinces and Territories protects 33.2% fewer species, which amounts to failing to protect 192 vertebrates, 721 plants, and 20 butterflies, of which 72 are considered at risk nationally, and 11 are considered at risk globally. The trade-off for prioritizing land independently within Ecozones is even more extreme, protecting 37.5% fewer species than the optimal

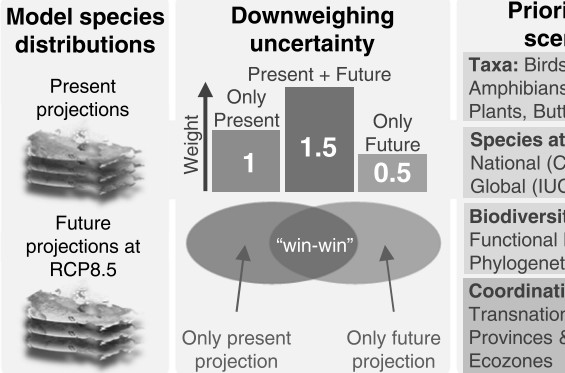

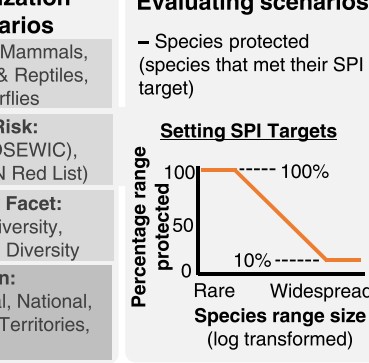

**Fig. 1 | Conceptual workflow of methods.** We used spatial prioritization to test whether prioritizing different dimensions of biodiversity versus prioritizing at different spatial scales matters more for 30×30 biodiversity outcomes. First, we build species distribution models to project the current and future ranges of all Canadian terrestrial vertebrates, plants, and butterflies. To incorporate climate change, we down-weighted future projections to account for uncertainty and to prioritize "win-win" areas of overlap between current and future ranges. Next, we designed 30×30 expansion scenarios that vary what dimension of biodiversity (i.e., taxa, species at-risk, facets) is prioritized as well as how protection is coordinated spatially. Finally, to evaluate spatial prioritization scenarios, we quantify both the amount of biodiversity captured using weighted endemism as well as the number of species protected based on a modified Species Protection Index (SPI), where a species is considered protected when it reaches or exceeds its species-specific conservation target.

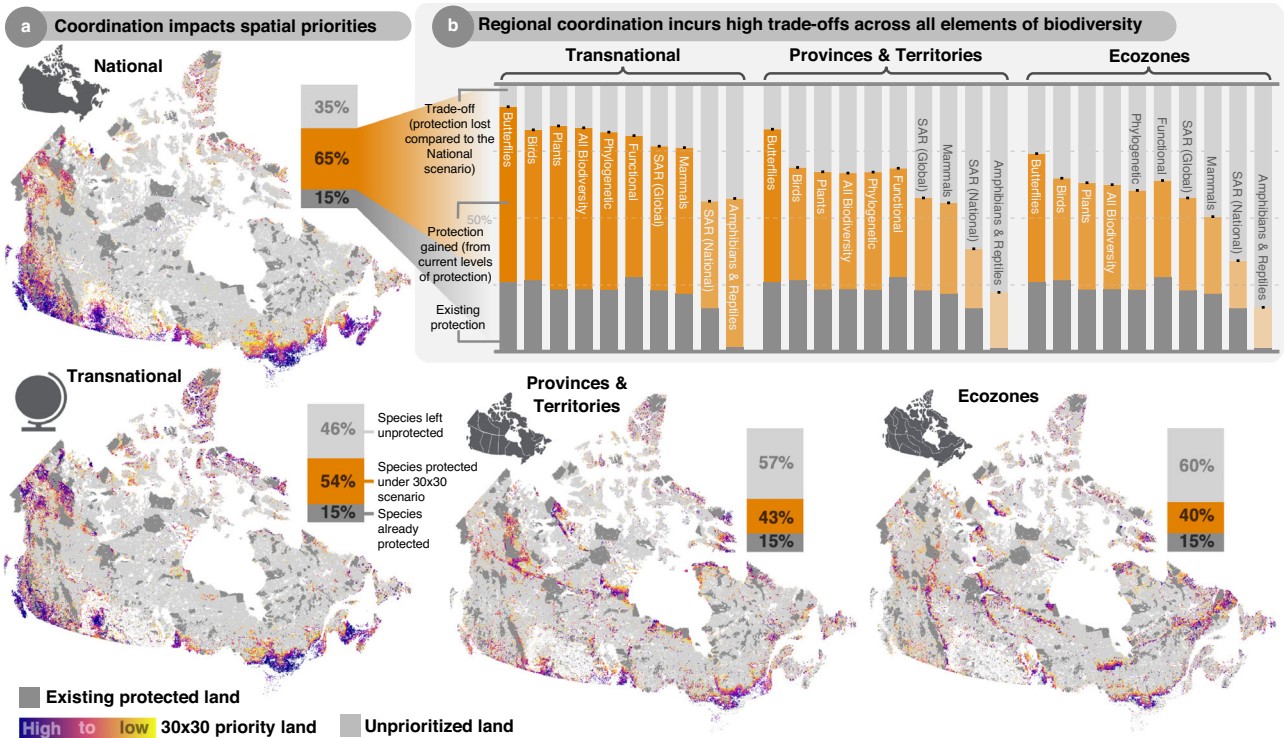

**Fig. 2 | Changing the spatial scale at which protection is coordinated impacts spatial priorities and biodiversity gains. a** Spatial priorities for the optimal *National* scenario as well as the *Transnational*, *Provinces & Territories*, and *Ecozones* scenarios are highlighted in color based on their priority rank. **b** These shifting spatial priorities incurred high trade-offs across all elements of biodiversity. Trade-offs represent the loss of potential protection from the optimal national scenario.

Trade-offs are calculated as the percentage difference between the amount of protection achieved by the alternative scenario and the amount of protection possible under the National scenario. For example, if the National scenario protects 80 species but the alternative scenario only protects 40 species, then the trade-off is −50% since the alternative scenario protects half as many species.

national scenario, which amounts to failing to protect over 1000 taxa. These decentralized regional approaches to reaching 30 × 30 are especially detrimental for the protection of amphibians, reptiles, and species at-risk (Fig. 2b), groups that are already neglected by existing protected land. For example, existing protected areas only protect 31 nationally listed species at-risk, only a single amphibian, and zero reptiles. And while nationally prioritizing land enables considerable conservation gains for these groups (protecting an additional 98 species at-risk and 53 amphibians and reptiles), regional approaches only protect a handful of additional taxa (21–26 additional species at-risk and only 8–11 additional amphibians and reptiles for Ecozones and Provinces & Territories priorities respectively).

Although national coordination enables the protection of Canadian biodiversity at large, it relies on highly uneven regional commitment, calling into question feasibility (Fig. 3). Specifically, most maritime Provinces (Nova Scotia, New Brunswick, and Prince Edward Islands), British Columbia, and the Yukon Territory in the Arctic, as well as coastal ecozones and those along Canada's southern border, contain a considerable amount of Canadian biodiversity and thus high priority land. As such, capturing Canada's biodiversity in protected areas will rely on coordination and cooperation across regions, with some jurisdictions protecting vastly more land than others. Interestingly, while regional priorities and spatial representation make capturing biodiversity significantly harder, transnational priorities (prioritizing species endemic to Canada) incur smaller trade-offs, suggesting that at least in Canada, national priorities can efficiently contribute to global goals and vice versa.

### Variation in spatial priorities across scenarios
Across all scenarios, spatial priorities varied greatly, resulting in only a small fraction (2.7%) of consistently prioritized land (Fig. 4a). These

areas, robust to different conservation priorities, contain ~13% of total biodiversity (based on weighted endemism) and ~23% of at-risk biodiversity, making them ideal candidates for protection. A surprisingly high portion of land (34.1%) was prioritized in only some scenarios, suggesting much of Canada's land conservation value is sensitive to shifting conservation priorities. This is due to the spatial mismatch between scenarios that prioritized different elements of biodiversity and scenarios that shifted the scale at which protection is coordinated, which is driven almost entirely by the regional scenarios that prioritize land independently across Provinces and Territories or Ecozones.

In general, prioritizing transnational and phylogenetic diversity best reflects the optimal *National* scenario (Fig. 4b), again supporting the idea that prioritizing biodiversity at national scales can serve global goals. To further explore similarities between scenarios, we visualized them in non-metric multidimensional space (NMDS), where *Ecozone*, *Province & Territory*, and *Amphibian & Reptile* scenarios were least similar to the *National* scenario (Fig. 5a). The axis that maximized variation in NMDS highlights the spatial differences between coordinating protection nationally versus transnational or regional approaches (Supplementary Fig. 2). To dig deeper into the spatial discrepancies between different priority scenarios, we directly compared scenarios that prioritize different taxonomic groups to those that vary the scale of coordination (Fig. 5b). This analysis revealed that the spatial dissimilarity between the two groups is largely driven by amphibian and reptile and regional priorities. However, unlike regional scenarios, where prioritizing land independently across Provinces and Territories or Ecozones results in severe reductions in our ability to safeguard biodiversity, prioritizing amphibians and reptiles incurs negligible trade-offs. Overall, uncoordinated regional protection hinders our ability to protect both biodiversity at-large as well as different elements of taxonomic, phylogenetic, and functional diversity

(Supplementary Data 1), indicating that coordinating protection across regions is far more important for the protection of biodiversity at-large than accounting for all taxonomic groups, species-at-risk, and biodiversity facets in the spatial planning process.

## Discussion

The recently adopted *Kunming-Montreal Global Biodiversity Framework* codifies the commitment of over 190 countries to protecting 30% of their terrestrial land by the year 2030[7]. And while protecting Earth's biodiversity is a global necessity and ultimately relies on uneven transnational cooperation[14], individual countries are responsible for reaching 30 × 30 within their territories and must make rapid strides to do so in less than a decade. Our results show that the success of 30 × 30 for biodiversity will likely depend on how individual countries coordinate protection strategies and identify spatial priorities. Assuming each country meets the target of 30% protected land, we find that a nationally coordinated strategy can both optimize the protection of each country's flora and fauna while also efficiently contributing to the protection of global biodiversity. On the other hand, uncoordinated regional approaches to reaching 30 × 30 could drastically limit the ability of individual nations to protect their biodiversity. In contrast, prioritizing different elements of biodiversity such as specific taxa, facets, or species at-risk only slightly impacted conservation gains, indicating that how nations choose to coordinate protection is vastly more important when it comes to protecting biodiversity at large, rather than accounting for all elements of biodiversity in spatial planning.

While reaching 30 × 30 will fundamentally depend on countless regional and local conservation initiatives, outcomes for biodiversity depend on how efficiently these initiatives contribute to the protection of biodiversity at large[2]. Our results show that nationally coordinated strategies can enable large conservation gains and efficiently serve global goals, while also benefiting regional conservation projects by providing broader context for local decisions, identifying opportunities for synergies (i.e., transboundary protected areas), assessing trade-offs, and facilitating progress tracking and reporting across scales[27]. These results align with past work that identifies stark differences in spatial priorities across scales as well as the benefits of broader scale approaches to coordinating the expansion of protected areas[14-16]. For many nations, the absence of a national strategy for 30 × 30 likely means significant reductions in our ability to safeguard biodiversity at large and especially species at-risk, amphibians, and reptiles. These species, largely neglected by existing protected areas, are particularly sensitive to further habitat loss, fragmentation, and climate change[28,29] and as such, representation in future protected areas may be critical for their persistence. Compared to transnational coordination, which would also benefit these taxa and optimally protect Earth's biodiversity in the broadest sense[14], national strategies are considerably more feasible, since coordinating within a nation is far easier than coordinating between them. This is especially true for areas of the world with contrasting economic, political, and socio-cultural conditions where nations are unlikely to collaborate on transboundary initiatives[17].

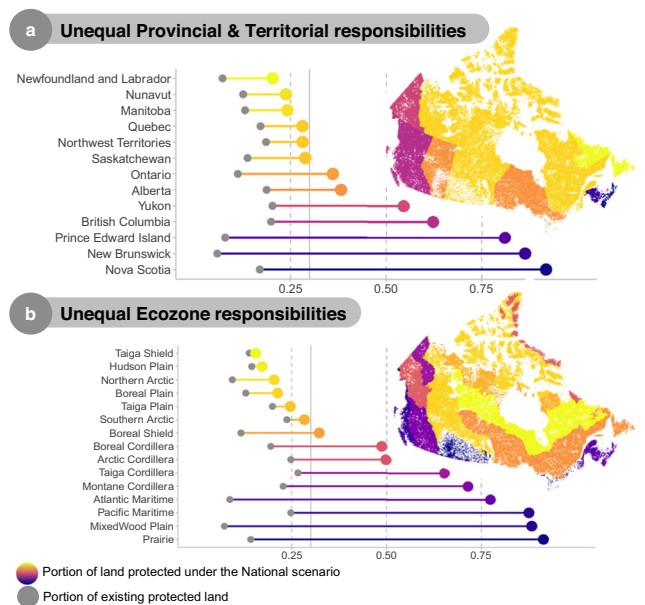

**Fig. 3 | Unbalanced regional responsibilities for protecting national biodiversity.** National coordination optimally protects Canadian biodiversity at large but relies on highly uneven regional commitment across Provinces and Territories (**a**) and Ecozones (**b**). Provinces and Territories and Ecozones are colored based on the portion of land included in the national 30 × 30 high priorities.

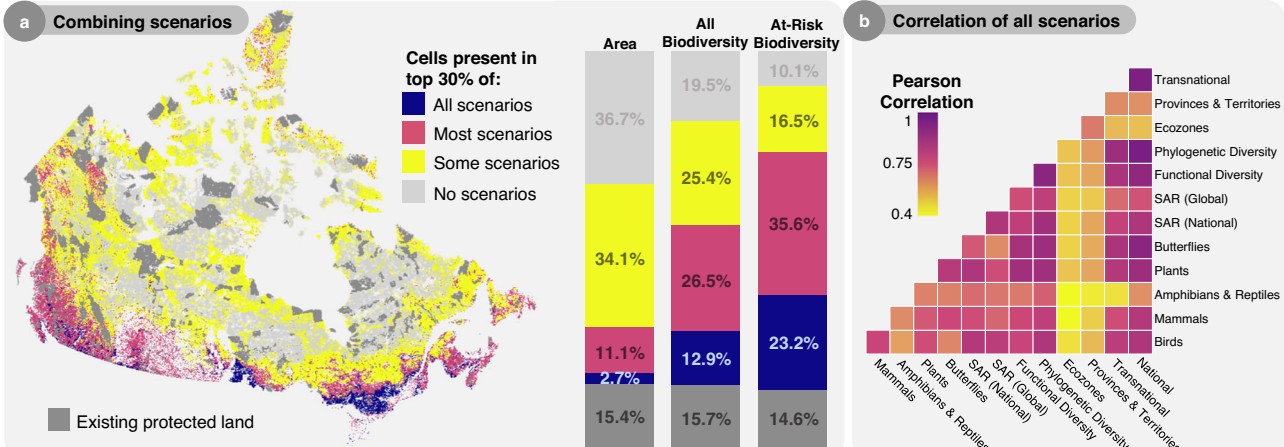

**Fig. 4 | Variation in spatial priorities across scenarios. a** Areas important for single or multiple priorities. Cells are colored based on their representation in the top 30% of cells across scenarios (All = present in all 13 scenarios, Most = present in at least seven scenarios, Some = present in less than 7, No = present in no scenarios). Bars represent the corresponding percent of land area, percent of total biodiversity, and percent of nationally listed at-risk biodiversity that is protected based on weighted endemism with each grouping of scenarios. **b** Correlations of the spatial overlap of each scenario where scenario pairs are colored based on the strength of Pearson correlation. For all cases, Pearson correlation tests produced significant *p* values (<0.05).

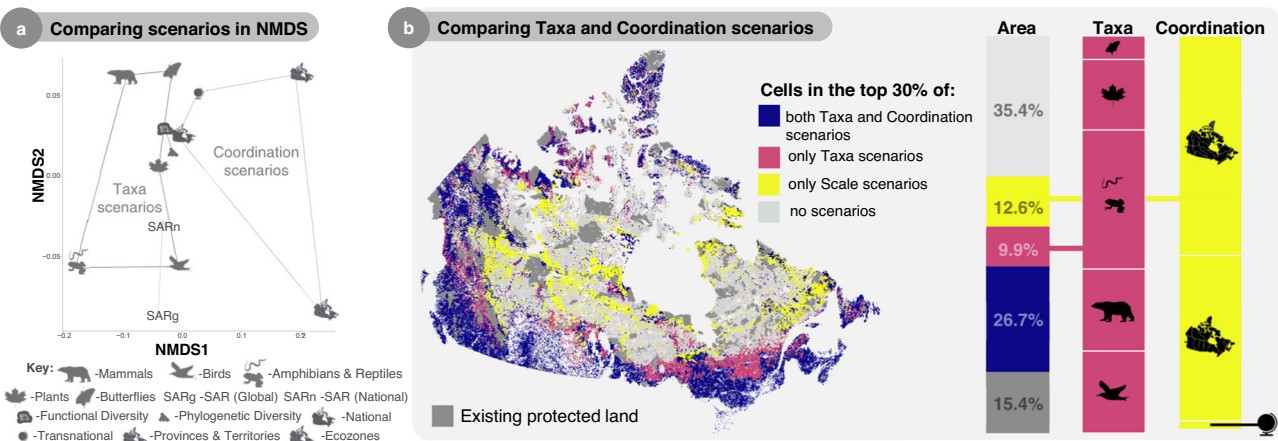

**Fig. 5 | Regional scenarios identify unique and dissimilar spatial priorities.** Comparison of scenarios that vary what taxa is prioritized versus how protection is coordinated across space by visualizing differences in: **a** nonmetric multi-dimensional space ($k = 2$, stress = 0.09) and **b** spatial overlap. In both panels, scenarios are represented by symbols indicating what was prioritized or how protection was coordinated. Colors represent land present in both taxa (i.e., Birds, Mammals, etc.) and coordination (i.e., National, Global, etc.) scenarios, only taxa, only coordination, or no scenarios. NMDS axes are visualized in Supplementary Fig. 2.

Alongside large conservation gains, national strategies for $30 \times 30$ can also facilitate spatial planning for climate resilience. Increasingly, evidence suggests that biodiversity in protected areas, despite being safeguarded from land use change, is still threatened by climate change[3]. This is especially true in northern regions like Canada, where climates are warming at multiple times the global average[22]. Unfortunately, recent work shows that over 50% of Earth's terrestrial protected areas are unlikely to adequately protect biodiversity into the future under climate change[30]. As such, incorporating climate forecasts into spatial planning for expanding protected areas to reach targets like $30 \times 30$ is a necessity. By prioritizing "win-win" areas which represent land that is climatically suitable today and into the future under climate change[24], national strategies can plan for predicted widespread climate change, while remaining robust to uncertainty in future climate models. Spatial planning for changing climates and shifting species distributions at national scales also enables cross-jurisdictional synergies like connectivity corridors or assisted migration programs and facilitates transnational cooperation to plan for transboundary range shifts[31]. Interestingly, our results contrast with past studies identifying important climate refugia in Canada[32,33], likely reflecting their focus on climatic stability of the land or predesignated refugia compared to our approach of climate resiliency at the individual species level. The high priority areas we identify depend on both endemism and climate resilience, thereby retaining complementarity and irreplaceability—core principles of systematic conservation planning.

Our findings suggest that national strategies for reaching $30 \times 30$ may hold the key to safeguarding nature into the future, but effective spatial planning relies on broadscale biodiversity modeling that overcomes existing data biases. While our ability to estimate the spatial distribution of biodiversity has greatly benefitted from recent increases in the accessibility and amount of species occurrence data[34], heavy data biases persist. Incomplete inventories and limited occurrence data can make modeling the distributions of rare, cryptic, or hard to observe species difficult, while spatial biases can distort model predictions leading to erroneous or misleading spatial projections[35]. Although increased sampling and cataloging of additional species observations can help alleviate these biases, rapid conservation action is necessary to prevent biodiversity loss and spatial planning cannot wait for these biases to be corrected at the observation level. Instead, it is up to researchers to develop workflows that overcome data limitations. Here, we demonstrate how spatial biases can be remedied to enable spatial planning even in nations with extremely biased data

landscapes such as Canada. We also show that even at fine resolutions spatial priorities and conservation outcomes are relatively insensitive to including or excluding different elements of biodiversity, suggesting that modeling all species or facets is not a prerequisite to effective spatial planning. Taken together, our findings suggest that by leveraging advances in biodiversity modeling and spatial planning, we can overcome existing data biases to design biodiversity-informed conservation strategies to combat current and future biodiversity loss.

Although the workflow demonstrated in this study can be applied to any nation to facilitate $30 \times 30$ planning, we chose to focus on just a single country. And while Canada comprises a land area comparable to Europe, contains 24% of our planet's remaining "intact" ecosystems[36], and has a critical role to play in the conservation of global biodiversity[37], more work is needed to evaluate whether our findings hold true for other nations. Canada's highly uneven distribution of biodiversity combined with a low portion of existing protected land may exacerbate the conservation gains possible under a nationally coordinated approach. Therefore, it is possible that nations with shallower biodiversity gradients and higher portions of existing protected land would not suffer the severe trade-offs we observed under regionally coordinated expansion scenarios. Nonetheless, our work demonstrates the power and importance of designing and evaluating different $30 \times 30$ scenarios at the national level to identify which choices and decisions significantly impact outcomes for nature. Additionally, while we choose to evaluate scenarios using SPI[25], our spatial prioritizations were not designed to optimize SPI but instead prioritized land based on endemism, complementarity, and balancing the protection of rare species against capturing the full range of biodiversity across the landscape. Future work to understand which indicators are sensitive to varying approaches to spatial planning would greatly benefit international targets like $30 \times 30$ by providing a robust indicator framework for nations to track and report progress[38].

While we address a broad range of biodiversity (species, clades, and multiple diversity facets), our prioritizations do not directly prioritize ecosystem services, connectivity, or equitable governance which are central to $30 \times 30$, nor do they account for the ability of habitat restoration projects to enhance biodiversity protection. Recent work is beginning to include these considerations[39–41] and explore the links between biodiversity facets and ecosystem services[42]. For example, phylogenetic and functional diversity, which is included in our analysis, can capture ecosystem services not typically represented in commonly used ecosystem service layers[43,44]. Nonetheless, more work

is needed to identify how spatial priorities for biodiversity contrast with other 30 × 30 priorities and design creative solutions that achieve positive outcomes for both nature and people.

Finally, Canada's path to reaching 30 × 30 and the establishment of new protected lands should advance Indigenous rights and title. Past work has demonstrated novel ways of designing protected areas that capture biodiversity and other 30 × 30 priorities and achieve equitable and just conservation outcomes through Indigenous consent, participation, and leadership[39]. This framework highlights both the need for and advantages of Indigenous Protected and Conserved Areas when it comes to meeting Canada's conservation and reconciliation goals. In the context of this work, these results fill an important gap in our understanding of how biodiversity is distributed across Canada and how the expansion of protected areas can maximize positive outcomes for nature—enabling decision makers to better consider biodiversity alongside other priorities when it comes to establishing new protected areas in Canada to reach 30 × 30.

In sum, our results confirm the importance of having biodiversity-informed national strategies for 30 × 30. Spatial planning that incorporates climate change, overcomes bias in biodiversity data, and accounts for uncertainty is both feasible and necessary[24]. Scalable biodiversity indicators are essential for understanding the trade-offs associated with different conservation priorities – including which species, functions, and lineages might be left unprotected[38]. In line with recent work demonstrating the need for national biodiversity monitoring[45], we find that nationally coordinated approaches to reaching 30 × 30 will not only most effectively protect each country's flora and fauna but can also efficiently contribute to global targets. On the other hand, nationally uncoordinated regional initiatives can limit the ability of area-based conservation to protect biodiversity at large[46]. Nonetheless, strong arguments can be made for conservation at local scales[47], and our findings do not invalidate the potential of such initiatives. Instead, we emphasize the importance of quantifying indicators, assessing trade-offs, and the urgent need for coordinated national strategies for reaching 30 × 30. The extent to which we can coordinate, cooperate, and share knowledge across scales and borders may well determine the success of international targets like 30 × 30 and consequently, the future of biodiversity on Earth.

## Methods

### Species lists and biodiversity data collection

Distribution data for all species was downloaded from the Global Biodiversity Informatics Facility[48]. We first curated species lists for each group. For plants, we used the database of vascular plants in Canada including only native species[49]. For butterflies, we downloaded from GBIF all observations recorded in Canada with an accuracy <5 km collected between 1990 and 2021 and combined this dataset with all observations available for the same time period in the eButterfly community science platform (www.e-butterfly.com). For terrestrial vertebrates, we first extracted all data from GBIF with a single locality recorded in Canada, then refined the list manually by examining each species removing non-native, extinct, and domestic species. We created an index using the "taxize" in package R[50] and manual refinement using a wide range of sources to crosscheck species names between the GBIF backbone taxonomy, IUCN range maps, traits, and phylogenetic data. We then extracted all occurrence points with a geographic location (excluding those with an unidentifiable geographic datum or location details). Due to the starkly different sampling intensities of some groups, we balanced having enough samples with the increased precision of geographic locations in more recent years. This meant that, for birds, we extracted data from 1990 to present and for all other groups, we extracted data from 1970 to present. We further filtered the bird occurrences to capture breeding ranges only (rather than the entire migratory range which often extends well outside of Canada). The occurrence data for birds were further filtered to include the

average start and end date of the breeding season (June to July), estimated for a random subset of 25% of species. For species whose data was limited by a lack of associated dates, we included all occurrence points but filtered by IUCN breeding polygons[51].

To further clean data points unlikely to represent the native distribution, we removed any points in core urban areas (i.e., areas designated 'urban or built-up' in the land-use/land cover data described below), which often included clusters of data points in zoos/sanctuaries. Then, we manually went through each vertebrate species to remove additional points outside of the known range of the species. To do this, we visually compared points to the IUCN range polygons, information about the specific species from guidebooks or online, and expert knowledge. In some cases, true outlying populations were outside of the range maps, and those populations were retained. For the species flagged as having outlier points, we then used an automated approach for removing outliers for flagged species by removing any points with an average distance to the nearest three other occurrence points of a certain distance in kms (distances established for each group of species independently). We used an average of three rather than one is that some outliers outside the native distribution were recorded multiple times spanning more than one grid cell. Following outlier removal, occurrence data was then gridded for a 1-km² grid on a Lambert Conformal Conic projection to match the climate data. Once gridded, data was thinned to only a single observation per grid cell, which left us with a list of grid cells in which each species had been observed, which was used as input into our models. All data were extracted from GBIF the week of 25 May 2021.

### Climate and edaphic explanatory variables

We used the following set of climatic variables that were biologically meaningful and had low correlation: mean annual precipitation (mm), chilling degree days (Degree days below 0 °C), precipitation as snow (mm), Hargreave's climatic moisture index and warming degree days above 18 °C. We used both current and future (2080 under RCP8.5) climate models from AdaptWest[52]. Current climate data is based on PRISM and WorldClim and spans 1991–2020. Future climate projections were downscaled from the Coupled Model Intercomparison Project Phase 6 based on an ensemble projection from 13 climate models and Representative Concentration Pathway 8.5 (RCP 8.5). RCP 8.5 is considered business-as-usual without climate change mitigation, which can produce the most extreme climate change projections. However, most discrepancy in SDM comes from the type of SDM rather than the GCP or RCP except for common species in no dispersal scenarios[53]. We are also discounting the future and truncating the distance that species can move in future projections (see below), so we felt this evidence justifies the ensemble GCM and RCP 8.5 scenario.

We also used topographic wetness index (calculated based on the 1-km digital elevation model using package "dynatopmodel" in R[54]), topographic ruggedness index (from Adaptwest), and an aggregated land cover layer based on MODIS land cover data and reprojected to our grid and reclassified to: unvegetated, hardwood forests, evergreen forest, mixed forests, shrubs, and grasslands[55]. For plants, we additionally used three variables to represent soil properties (topsoil silt fraction, subsoil pH, and topsoil organic C content) from the Unified North American Soil Map[56] (0.25 degree resolution) that were projected to match the 1-km² climate raster.

### Species distribution models

We used a set of species distribution models that performed well in preliminary tests (variable importance, realistic response curves, visual checks of realistic mapped predictions, and ability to handle interactions between categorical and continuous variables) for a variety of different organisms. All organisms had strongly biased GBIF sampling with southern latitudes being much better sampled. We addressed this sampling bias by (1) gridding and thinning occurrence data so that a

species is either present in the grid cell or not (based on a single occurrence for all non-bird species and two occurrences for birds) and (2) accounting for this bias in different ways when fitting models. Furthermore, we randomly selected presences for wide ranging species to limit the effect of spatial autocorrelation.

We fitted models with two separate algorithms (Boosted Regression Trees; BRT, and Maximum Entropy; MaxEnt), which typically have strong predictive power[57–59]. We used two bias correction methods. For BRTs, we fitted models with all environmental predictors plus sample effort (all GBIF observations of plants and vertebrates within a 30-km$^2$ surrounding area) and Human Footprint Index (HFI). Then, we set the sample effort to its maximum value for prediction. MaxEnt uses a target background approach to account for bias[60]. All presences were used in the model unless they exceeded 5000, in which case 5000 presences were randomly drawn along with 10,000 absences. While this means there is a presence: absence imbalance for rare species, it was necessary to fully cover the environmental space as we were projecting each species across the study area. Importantly, we are not comparing across species (e.g., calculating richness), Zonation 5 (and the weighted endemism metric more generally) scales all species individually. Therefore, the magnitude of the habitat suitability of one species does not need to be compared to the others. Species with fewer than ten presences were excluded, as were species whose models did not fully converge. Models were fitted using the "dismo" package in R[61] on data from all of the continental United States and Canada to avoid environmental truncation and predicted to a 5-km$^2$ grid after initial checks to verify that predicted ranges were very similar between resolutions. BRTs were fit using the following settings: family = bernoulli, tree complexity = 4, learning rate = 0.001, bag fraction = 0.6. MaxEnts were fit using only hinge features to avoid overfitting with the following settings: args = c("-P","noautofeature", "nolinear", "noquadratic", "nothreshold", "noproduct").

Model validation is a major challenge when true absences are lacking and particularly when we know that input data is strongly biased along the environmental gradients from which we are fitting the model. Our main concern was accounting for this discrepancy, so we validated models based on a set of three comparison extents: Canada-wide, colder regions (defined as 80% of the land with the largest Chilling degree days) and warmer regions (the top 20% with the smallest Chilling degree days). We calculated area under the receiver operator characteristic curve and area under the precision-recall curve. We did not calculate metrics that require a single threshold, as we used the numeric values directly in Zonation. Summarized validation scores are available in Supplementary Table 1.

### Model projections

We predicted to both present and 2080 (under RCP8.5) climatic conditions at 5-km$^2$ resolution for all of Canada. We acknowledge that by using only a single ensemble GCM we are minimizing some differences between GCMs for later time periods[62], however, our use of downweighed future projections makes their contribution to spatial prioritization limited to mainly helping define "win-win" areas of climate stability and so using different/multiple GCMs would likely not change the surprisingly stable high spatial priority regions in Canada. Model projections were then downscaled to 1-km$^2$ resolution to match the other input layers for Zonation. Instead of using four inputs per species (BRT current, Maxent current, BRT future, and Maxent future) we averaged BRT and Maxent models to make a single current and a single future projection using the following weighting scheme: BRT = 0.7, Maxent = 0.3. These weights were chosen based on our ability to incorporate stronger bias correction into the BRTs and because ensemble models generally have the strongest predictive power[63]. Because bias correction and climate change forecasts can lead to unrealistic long-distance shifts in species in species distributions, we restricted vertebrate distributions to a maximum of 500 km beyond

their IUCN range polygon. For species that did not have IUCN polygons we created polygons by roughly tracing expert estimated range maps found online through either NatureServe (https://www.natureserve.org/) or governmental websites or by tracing around the boundaries of GBIF observations if no range map could be found. We hand validated all current SDM projections for all vertebrates, 10% of plants, and as many butterflies as possible by comparing SDM range maps against expert range maps from field guides and a variety of online sources such as NatureServe. Where necessary we also consulted experts to determine the validity of model predictions to ensure accuracy.

### Accounting for climate change and uncertainty

Including future species distributions can enable spatial planning to incorporate climate change, but also introduces various sources of uncertainty. To overcome this, we used a weighting scheme to prioritize "win-win" areas in Zonation illustrated in Fig. 1[24]. This approach assigns the highest weights to cells present in both current and future distributions, since these are the areas for which uncertainty in species presence (today and into the future) is the lowest. The cells with the lowest weights are those in only the future distribution since the potential of species to occupy those cells is highly uncertain. This approach allows us to identify areas within species current ranges that are most likely to act as climate refugia, while retaining the principles of irreplaceability and complementarity central to systemic spatial planning.

### Mask layers for Zonation

Along with the biodiversity inputs, Zonation requires a few other layers. To include existing protected areas and insure those are prioritized first, we used a hierarchical mask layer. We identified existing protected areas using the Canadian Protected and Conserved Areas Database[21]. Polygons were rasterized and projected to the 1-km$^2$ climate grid, and cells with at least 43% of their area within protected areas were considered protected. We chose to include other effective conservation measures (OECMS) since Canada counts OECMs towards international targets. The 43% threshold was set so that the total number of protected cells would be roughly the same portion as the total amount of protected land (13.5% when including areas of high human footprint and ceded Indigenous land).

Zonation also accepts a base mask layer that delineates the study area. We wanted to focus our analysis on land in Canada that could feasibly house future protected areas. As such, we excluded areas of high human footprint (representing urban, agricultural, industrial, or other high disturbance areas). We identified areas of high human impact using the recently published Canadian cumulative Human Footprint Index[20] reprojected to 1-km$^2$. Following their approach, we considered any cell with an HFI value above 10 to represent an area of "high human footprint", representing roughly 5.7% of Canada, and excluded it from our analysis. The remaining cells represent largely intact "ecosystems" and thus good candidates for protection. Because Canada is largely situated on unceded Indigenous land, we chose to acknowledge what Indigenous land has been ceded by excluding it from our analysis, since Indigenous land has been shown to contain levels of biodiversity similar to what is observed in protected areas[64], and the government has no jurisdiction to establish new protected areas on Indigenous land. To identify Indigenous land, we used the Aboriginal Lands of Canada Legislative Boundaries Database[65], rasterized to 1-km$^2$ resolution. We used the same threshold (43%) to identify "Indigenous" cells, representing roughly 6% of Canada.

### Expansion scenarios design

To assess how different conservation priorities impact our ability to protect biodiversity, we designed 13 conservation scenarios. The national scenario, which represents the optimal scenario for Canada, prioritizes all species simultaneously, weighted equally across

kingdoms (so that vertebrate, plant, and butterfly diversity each receive the same total weight). To assess how the inclusion or exclusion of specific taxa impacts spatial priorities, we prioritized birds, mammals, amphibians & reptiles, plants, and butterflies separately. We also designed two scenarios to prioritize species-at-risk. We used two species at-risk assessments, the Committee on the Status of Endangered Wildlife in Canada (COSEWIC) assessment and IUCN's Red List, which correspond to national and global assessments. Since, at a national scale, countries are more likely to use their own assessments, we reported the COSEWIC results as the main species at-risk results in the text. For these scenarios, we only included species listed as threatened, endangered, or special concern in the case of COSEWIC and vulnerable, endangered, and critically endangered for IUCN.

To assess conservation scenarios that prioritize functional and phylogenetic biodiversity facets, we weighed species according to their functional or phylogenetic distinctiveness. To calculate functional distinctiveness, we used hypervolume contribution scores, where we built a single hypervolume for vertebrates, plants, and butterflies separately, and calculated the contribution of each species. Contributions were then standardized so the sum of all contributions equaled 1 for vertebrates, plants, and butterflies separately, thus weighing each kingdom evenly during Zonation. To build hypervolumes, we used two principal coordinates from a distance matrix calculated from normalized functional traits[66] in the 'BAT' package for R[67]. For vertebrates, we used diet, body mass, litter clutch size, generation length, lifespan, wintering strategy, and age at sexual maturity. Vertebrate functional trait data was sourced from various databases, including the Amniote database[68], AmphiBIO database[69], PanTHERIA database[70]. For plants, we used seed mass, height, specific leaf area, lifespan, nitrogen fixation capacity, growth form, photosynthetic pathway, dispersal syndrome, reproductive timing, leaf compundness, and woodiness, all downloaded from the TRY database[71]. For butterflies we used expertly estimated mobility, wingspan, range size, and larval host plant breadth[72] combined with the recently compiled LepTraits database[73]. To fill gaps in the trait data, we imputed missing values using phylogenetic vector regressions (PVRs) in the "PVR" package for R[74] calculated from phylogenetic trees to aid random forest imputation in the "MissForest" package for R[75]. To calculate phylogenetic distinctiveness, we used existing phylogenetic trees for vertebrates[76–79], plants[80], and butterflies[72]. After pruning trees to only include Canadian species, we calculated distinctiveness using the "evol_distinct" command in the "phyloregion" package for R[81].

To assess how spatial scale of coordination impacts spatial priorities, we used three scenarios. The *Transnational* scenario prioritizes global biodiversity by weighing species in Zonation based on their Canadian (weighted) endemism (i.e., the portion of their North American range found in Canada). The *Provinces & Territories* scenario protects an equal portion of each province and territory, so protected areas are spread evenly across the political landscape. The *Ecozones* scenario protects an even portion of each ecozone, achieving even spatial representation from an ecological point of view. Both *Provinces & Territories* and *Ecozones* scenarios represent regional scale priorities. All scenarios use the same Zonation 5 settings outlined in the following paragraph.

## Spatial prioritization

To prioritize land in each conservation scenario, we used Zonation 5 with CAZ2 marginal loss, which balances priorities across species and is new in Zonation 5[23]. We chose to exclude areas of high human footprint and Indigenous land from our analysis, as these represent areas where the establishment of new protected land is unlikely (i.e., due to high costs or low availability) or not appropriate. As such, although Canada has protected 13.5% of its terrestrial land, that represents 15.4% of land included in our analysis. We used a hierarchical mask layer in Zonation which allows for the initial

prioritization of existing protected land, before prioritizing remaining cells, allowing for complementarity in spatial planning. For *Provinces & Territories* and *Ecozones* scenarios, we used subregions, where a full prioritization was performed for each subregion separately, and the prioritizations were stitched together in one final raster, representing all of Canada. This allowed each subregion to prioritize the species and endemism specific to that subregion alone, accounting for already protected land, and enabling complementarity. The output of Zonation runs is a final raster for all unmasked cells, ranked according to their priority.

## Reaching 30 × 30 targets

From these rank maps, we simulated 30 × 30 by selecting the top 30% of cells in each scenario, including already protected areas. While all rank maps are available in the supplementary information (Supplementary Fig. 1), we chose to only include the national prioritization as well as the scale of coordination scenarios in the main text (Fig. 2a)

## Biodiversity trade-offs

To assess the biodiversity trade-offs associated with different conservation priorities, we used a modified SPI to measure the percentage of taxa considered adequately protected[25]. SPI works by setting species-specific protection targets, based on how common or rare a species is across the landscape. Since we used probabilistic species distributions, we consider a species range to be the sum of probabilities in all cells across a landscape. The top 10% most common species, with large ranges, require at least 10% of their range inside protected areas to be considered adequately protected. The top 10% rarest species, with small ranges, require 100% of their range inside protected areas to be considered adequately protected. For the 80% of species between these thresholds, we used a log-linear model to set species-specific goals. Species that met or surpassed their conservation goals were considered "protected", while species that did not meet their goals were left "unprotected" by conservation scenarios.

We calculated biodiversity trade-offs as the relative difference between the percentage of species considered "protected" under the *National* prioritization scenario and the number of species considered "protected" under other scenarios. For example, if the national prioritization protected 10 species, and a different conservation priority scenario only protected five species, the biodiversity trade-off would be (5/10)*100 = 50%. Or in other words, the different conservation priority protects 50% of the biodiversity compared to the optimum national prioritization. We calculated biodiversity trade-offs for all biodiversity (all species) as well as for birds, mammals, amphibians & reptiles, plants, butterflies, COSEWIC species at-risk, and IUCN species at-risk separately. In addition, we also quantified functional and phylogenetic biodiversity trade-offs by calculating the total functional and phylogenetic contributions of "protected" species as a percentage of total Canadian functional and phylogenetic diversity (the sum of all species contributions).

We also calculated trade-offs using weighted endemism which represents the "percentage" of biodiversity in protected cells. Because SPI and weighted endemism trade-offs were highly correlated, we chose to report SPI in the main text (Fig. 2b) and include both in the supplementary information (Supplementary Data 1).

## Spatial commitments

To highlight the uneven challenge posed when prioritizing biodiversity, we calculated the total amount of each Province & Territory and Ecozone in the top 30% of the national prioritization scenario (Fig. 3).

## Comparing priority scenarios

To compare scenarios, we used multiple methods. First, we visualized overlap between 30×30 expansion scenarios by highlighting cells in

the top 30% of all scenarios, most scenarios (i.e., seven or more), some scenarios (i.e., six or fewer), and no scenarios (Fig. 4a). Then we calculated the pairwise Pearson correlations for all scenarios and visualized them using a heatmap (Fig. 4b). Strong correlations represent spatially similar prioritization scenarios, whereas weaker correlations represent spatially divergent scenarios. To compare scenarios further, we use nonmetric multidimensional scaling in the 'vegan' package for R[82], treating each prioritization as a site, and each cell as a species (Fig. 5a). NMDS revealed two distinct spatial axes of variation between all scenarios, achieving a low stress value of 0.09. Finally, to highlight the spatial differences between clade and political scenarios, we once again visualized overlap, this time quantifying which specific scenarios were driving the spatial dissimilarity between the two groups (Fig. 5b).

### Reporting summary
Further information on research design is available in the Nature Portfolio Reporting Summary linked to this article.

## Data availability
The range polygon data, as well as the functional, phylogenetic, and transnational weights, all scenario rank maps, and generated in this study have been deposited in the FigShare database under accession code https://figshare.com/s/0551e56687ba119c7bb8.

## Code availability
The code used to make the figures are available in the following repository: https://figshare.com/s/0551e56687ba119c7bb8.

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

## Acknowledgements

The authors would like to acknowledge that while they were able to account for ceded Indigenous land in their analysis, much of what we now call Canada remains unceded, rich with a history of Indigenous oppression. As such, the establishment of new protected areas and Canada's path to 30×30 should involve Indigenous communities, knowledge, and perspectives so as to advance Indigenous rights and title. Additionally, the authors would like to thank Darren Li and Cole Lee for their work in gathering the functional trait data, Stefano Mammola for his help with computing functional hypervolumes, and Abbie Gail Jones and Olivia Rahn for their many helpful comments throughout the duration of the project. This work was supported by NSERC DG grant RGPIN-2019-05771 (L.J.P.) and by a NSERC CGS-D award (I.E.).

## Author contributions

I.E. and L.J.P. conceived the idea for the manuscript. I.E., A.B., D.C., F.R., and L.J.P. developed and executed the methodology. Results were generated by I.E. and L.J.P. and visualized by I.E. I.E. wrote the first draft of the manuscript and all authors provided critical feedback. L.J.P. supervised the project and acquired the funding.

## Competing interests

The authors declare no competing interests.
