## [Peer Review File · Nature Communications]

30x30 biodiversity gains rely on national coordinationREVIEWER COMMENTS

Reviewer #1 (Remarks to the Author):

This manuscript analyzes the distribution of conservation priority areas for biodiversity at national and subnational scales in Canada. Its major strengths are the multi-scale comparison, the analysis of overlap between current and projected future climate-based ranges for a taxonomically diverse set of taxa, and comparison of multiple biodiversity-related metrics including functional diversity. Its major limitations are inadequate comparison against previous studies with similar aims and scope, and (in my view) inaccurate framing of the findings comparing national and subnational priority areas.

In terms of placing in context of previous work, this includes both multi-scale comparisons in other regions (e.g. Zhu 2021 10.1126/sciadv.abe4261), and 30x30-related analyses that encompass Canada. This could include Stralberg 2020 (10.1111/conl.12712) which analyzes refugia, connectivity etc. and Saunders 2023 (10.1002/fee.2592) which analyzes refugia (and interestingly, identifies priority areas that contrast with those of this manuscript). Also, some Canada-specific studies have appeared (Currie 2023, 10.1111/csp2.12924).

The second limitation is more important and central to the current framing of the manuscript. The manuscript effectively focuses on one element of biodiversity (species-related), although it appropriately considers both taxonomic and functional diversity at that level. The 30x30 target, as found in the recently adopted KM-GBF, is framed as designed to protect "areas of particular importance for biodiversity and ecosystem functions and services, are effectively conserved and managed through ecologically representative, well-connected and equitably governed systems". This tiers from the multi-faceted definition of biodiversity in KM-GBF Goal A, which includes not only the goal of halting human-induced species extinctions, but also ensuring that "The integrity, connectivity and resilience of all ecosystems are maintained, enhanced, or restored, substantially increasing the area of natural ecosystems". Also, conservation of genetic diversity, which often requires a focus on regional populations as well as on conserving the species range as a whole. Currently, the manuscript effectively states "species-level biodiversity is the key goal, and subnational prioritization strategies are inefficient at this and thus suboptimal", rather than stating "strategies focused on species-level biodiversity identify different priority areas than do strategies focused on other elements of biodiversity (or other goals such as ecosystem services), how can we resolve this in planning?".

Practically-speaking in Canada, species-richness based prioritizations will identify areas in the warmest and most speciose portion of the country along its southern edge, and concentrating new conservation areas will achieve certain goals at the expense of others (e.g. conservation of large intact ecosystems).

Some of the methods that the authors criticize such as Ecozones may or may not be optimal, but were designed in part to achieve the "ecologically representative" and ecosystem-level conservation goals stated above. Therefore framing such as [lines 147-8] uncoordinated regional protection vastly hinders our ability to protect biodiversity" are not acknowledging the full context of biodiversity and other goals. I fully agree with the authors as to the necessity for national (as well as global and subnational) strategies, but in my view their interpretation of their specific findings is poorly contextualized.

Minor point: the study excluded 6% of Canada that they categorize as Indigenous held areas (it should be noted that this is a small fraction of areas which First Nations consider as traditional lands and have proposed conservation and management plans for). I acknowledge that this step was well-intentioned, and at 6% likely does not affect results greatly. However, the Canadian government is centering its 30x30 strategy around supporting Indigenous-led conservation, and many of the proposed new areas are Indigenous Protected and Conserved Areas. So it would be more useful to explain how this work could support IPCA planning rather than (or in addition to) exclude Indigenous held areas from the analysis.

Reviewer #2 (Remarks to the Author):

Thank you for the opportunity to review the manuscript, "30x30 biodiversity gains rely on national coordination". I found this paper interesting and well written. The noteworthy results are that how we coordinate protection at different spatial scales has more implications on how we protect biodiversity than the choice of which element of biodiversity to prioritize. Because of this, I do think this work presents novel elements that can help guide conservation efforts around area based conservation targets and would be a good addition to the literature.

I find the results provocative, that national coordinate efforts will provide better biodiversity protection- however, this does leave unanswered questions that the authors could try to address. Such as- how can a national government coordinate an effort such as this at the regional and local scale in a way that may appear to be uneven and/or inequitable? How can this be achieved when regional/state/province and local protected areas are often acquired based on availability, willingness of the landowner to sell, cost, and other local factors? The authors note that in Canada, most protected areas are spread out evenly across the political landscape (can you please provide citations for this?), however this is not the case for many countries. Do the authors have suggestions on how to implement this?

One concern I have is that the framing makes it seem that national is best when the national and regional approaches are actually prioritizing different things. The national scenario weighs all taxa evenly when looking to prioritize areas across the landscape, so by the nature of how it is set up will try to maximize its efficiency in selecting areas that are representative across all taxa. It is not surprising that they do well in capturing biodiversity. On the other hand, the regional approaches either weigh species by endemism or try to spread out priority areas evenly or to achieve ecosystem representation. In these cases, some species will be prioritized higher (so less balanced across all taxa), or the output be spread out the areas across a more diffuse network to capture many areas but not necessarily the best possible configuration. These are all very different approaches to prioritization that have different principles at the root of them, so it is not surprising they are selecting different areas. I think the authors should be more upfront about what national and regional coordination means, and that they are based in different approaches. National approaches could also focus on endemism or spreading out protected areas across the country, so its more about how the effort is coordinated than who is doing it. I would love to see the text updated to reflect this nuance. I was surprised to find it buried in the methods.

I see in the methods that areas of high human footprint and Indigenous lands were excluded from the analysis. I think you need to say this early on in the introduction/results as I was thinking about this as I read the paper. There is no mention or assessment of how 30x30 could also be used to improve human well-being and nature access, so excluding areas of high human footprint would exclude restoration efforts that could benefit both people and nature. Perhaps some discussion on how this paper is focused on more strict protections under 30x30, rather than other conservation efforts that may count under areas based conservation targets. There are criticisms on area based targets on how they overlook human communities, where we could be finding solutions that co-benefit both. So although avoiding these areas in the analysis helps avoid issues of how protection has led to historical conservation legacies of land displacement, it also avoids the conversation that working in co-production with communities and Indigenous Peoples and Local Communities (IPLC) can be a pathway towards 30x30 as well. In relation to this, the term "wilderness" also has colonial roots and has been considered problematic to some.

I also would like more exploration of the limited amount of overlap between the spatial priority scenarios, only 2.8%, with only 34.2% of land prioritized in only some scenarios. This part was glazed over quickly, stating that it was "sensitive to shifting conservation priorities". It would be great to know why there is little consistency in prioritized land across scenarios, and perhaps and understanding of the implications of this.

SPecies Distribution models:

I have a few comments on the underlying species distribution models that require more information

1) It seems that all species were trained on the same geographic extent. If so, this can have serious implications for model outputs, especially for more range restricted species. The spatial extent of the training region should be tailored to the species realistic geographic space. See citations below

(VanDerWal, J., Shoo, L. P., Graham, C., & Williams, S. E. (2009). Selecting pseudo-absence data for presence-only distribution modeling: how far should you stray from what you know?. *Ecological Modelling*, 220(4), 589-594.)

(Saupe, E. E., Barve, V., Myers, C. E., Soberón, J., Barve, N., Hensz, C. M., ... & Lira-Noriega, A. (2012). Variation in niche and distribution model performance: the need for a priori assessment of key causal factors. *Ecological Modelling*, 237, 11-22.)

2) Did you do any filtering of the occurrence data to account for spatial autocorrelation, bias, and model overfitting?

3) I don't see information on model testing- Did you run any cross-validation or similar testing and if so, did you use any geographic data partitioning for model assessment?

4) What was the reasoning for averaging the BRT and Maxent and in the proportions that you did? Please provide more information and/or citations.

Minor comments-

Please add the time period for future climate change along with the RCP around line 331 (I note it is mentioned later, but would be good to have included with the data description).

Line 33- there is mention that SDM output discrepancy comes from the SDM rather than the GCP or RCP. However, there is much divergence in climate scenarios across GCMs after the 2050 time period. Given you are using an ensemble and looking at 2080, your model outputs will mute some of these differences. There is support in the literature that using ensemble methods may not be representative of real conditions, and that a range of GCMs should be used to get a sense of prediction uncertainty (see Buisson, L., Thuiller, W., Casajus, N., Lek, S., & Grenouillet, G. (2010). Uncertainty in ensemble forecasting of species distribution. *Global Change Biology*, 16(4), 1145-1157.). At the least the authors should acknowledge this if they are not able to present information on uncertainty of their model outputs.

SDMs

Were all the default settings used in Maxent and BRT using the dismo package? please provide more details so that the modeling methods can be fully assessed and/or repeated.

Detailed Comments and Responses

Response to comments 1-2 from editor are available in the cover letter.

Reviewer Comments

Comment 3:

Reviewer #1 (Remarks to the Author):

This manuscript analyzes the distribution of conservation priority areas for biodiversity at national and subnational scales in Canada. Its major strengths are the multi-scale comparison, the analysis of overlap between current and projected future climate-based ranges for a taxonomically diverse set of taxa, and comparison of multiple biodiversity-related metrics including functional diversity.

Response 3: Thank you for agreeing to review our manuscript and for providing thoughtful and helpful comments. We are pleased that you found novelty in our work and are happy to provide an edited manuscript that incorporates all your suggestions.

Comment 4:

Its major limitations are inadequate comparison against previous studies with similar aims and scope, and (in my view) inaccurate framing of the findings comparing national and subnational priority areas.

Response 4: Where appropriate we have added additional information that compare our study to previous studies (including the ones you suggested). We believe this has strengthened the introduction and rationale of our manuscript and has helped communicate the novelty of our approach and our findings and has better placed our work in the context of the broader goals and values of systematic conservation planning. We have adjusted the way we frame our results to represent our approach more accurately as per your suggestions below.

For example, we add additional citations/explanations of previous research in the introduction and discussion, such as:

L49-50: We reference Zhu et al. 2021 in the introduction for an example of how spatial priorities and biodiversity outcomes are impacted by coordination and come back to this reference in the discussion (L180-182)

L206-211: "... our results contrast with past work identifying important climate refugia in Canada (Sanders et al 2023, Stralberg et al 2020)..."

Comment 5:

In terms of placing in context of previous work, this includes both multi-scale comparisons in other regions (e.g. Zhu 2021 10.1126/sciadv.abe4261), and 30x30-related analyses that encompass Canada. This could include Stralberg 2020 (10.1111/conl.12712) which analyzes refugia, connectivity etc. and Saunders 2023 (10.1002/fee.2592) which analyzes refugia (and interestingly, identifies priority areas that contrast with those of this manuscript). Also, some Canada-specific studies have appeared (Currie 2023, 10.1111/csp2.12924).

Response 5: We have referenced all your suggestions in the main text to better place our study in the broader field of spatial prioritization conservation research, both globally and in Canada (see details in Response 4).

Comment 6:

The second limitation is more important and central to the current framing of the manuscript. The manuscript effectively focuses on one element of biodiversity (species-related), although it appropriately considers both taxonomic and functional diversity at that level. The 30x30 target, as found in the recently adopted KM-GBF, is framed as designed to protect “areas of particular importance for biodiversity and ecosystem functions and services, are effectively conserved and managed through ecologically representative, well-connected and equitably governed systems”. This tiers from the multi-faceted definition of biodiversity in KM-GBF Goal A, which includes not only the goal of halting human-induced species extinctions, but also ensuring that “The integrity, connectivity and resilience of all ecosystems are maintained, enhanced, or restored, substantially increasing the area of natural ecosystems”. Also, conservation of genetic diversity, which often requires a focus on regional populations as well as on conserving the species range as a whole.

Currently, the manuscript effectively states “species-level biodiversity is the key goal, and subnational prioritization strategies are inefficient at this and thus suboptimal”, rather than stating ‘strategies focused on species-level biodiversity identify different priority areas than do strategies focused on other elements of biodiversity (or other goals such as ecosystem services), how can we resolve this in planning?’.

Practically-speaking in Canada, species-richness based prioritizations will identify areas in the warmest and most speciose portion of the country along its southern edge, and concentrating new conservation areas will achieve certain goals at the expense of others (e.g. conservation of large intact ecosystems).

Some of the methods that the authors criticize such as Ecozones may or may not be optimal, but were designed in part to achieve the “ecologically representative” and ecosystem-level conservation goals stated above. Therefore framing such as [lines 147-8] uncoordinated regional protection vastly hinders our ability to protect biodiversity” are not acknowledging the full context of biodiversity and other goals. I fully agree with the authors as to the necessity for national (as well as global and subnational) strategies, but in my view their interpretation of their specific findings is poorly contextualized.

Response 6: This is a compelling point and comment, and we agree that our manuscript does not address other interpretations of Target 3 in the KM-GBF. As per your comment, we have adjusted the overall framing of our manuscript to better reflect our focus on species-level biodiversity and to place our “critic” of sub-national strategies more accurately within the goal of maximizing positive outcomes for biodiversity. We have also added a section in the discussion that acknowledges our decision to not include ecosystem services directly or ecological connectivity in this prioritization (L249-258). That said, our prioritizations and analysis include a broad scope of biodiversity, since we also quantify and assess the trade-offs of regional coordination for the protections of different element of biodiversity (i.e., different taxonomic groups or species at risk) as well as phylogenetic and functional diversity.

Comment 7:

Minor point: the study excluded 6% of Canada that they categorize as Indigenous held areas (it should be noted that this is a small fraction of areas which First Nations consider as traditional lands and have proposed conservation and management plans for). I acknowledge that this step was well-intentioned, and at 6% likely does not affect results greatly. However, the Canadian government is centering its 30x30 strategy around supporting Indigenous-led conservation, and many of the proposed new areas are Indigenous Protected and Conserved Areas. So it would be more useful to explain how this work could support IPCA planning rather than (or in addition to) exclude Indigenous held areas from the analysis.

Response 7: We thought a lot about the best action to take regarding Indigenous held areas in our study. One major issue in Canada is that ceded Indigenous land represents a small fraction of the traditional and claimed lands of Indigenous groups, so there is no perfect way to fully account for land rights, claims, or ownership. Our rationale for excluding ceded areas from our analysis was to firstly acknowledge that government does not have jurisdiction over these lands and secondly to avoid imposing our western concept of “spatial priority” on land that is not ours and whose stewards may not agree with our approach to prioritizing land. We agree that IPCAs are important and central to the Canadian government’s path to reaching 30x30, however since IPCAs inherently invoke a different set of values compared to non-Indigenous PAs, informing future IPCAs is well beyond the scope of our manuscript. That said, we have added a section to the discussion about the importance of IPCAs and Indigenous perspectives to 30x30 in Canada and have referenced past work focused on proposing new IPCAs (L259-268).

Comment 8:

Reviewer #2 (Remarks to the Author):

Thank you for the opportunity to review the manuscript, "30x30 biodiversity gains rely on national coordination". I found this paper interesting and well written. The noteworthy results are that how we coordinate protection at different spatial scales has more implications on how we protect biodiversity than the choice of which element of biodiversity to prioritize. Because of this, I do think this work presents novel elements that can help guide conservation efforts around area based conservation targets and would be a good addition to the literature.

Response 8: Thank you for agreeing to review our manuscript and we are glad you found novelty in our work. Your comments were helpful, and we managed to incorporate all of them into the revised document.

Comment 9:

I find the results provocative, that national coordinate efforts will provide better biodiversity protection- however, this does leave unanswered questions that the authors could try to address. Such as- how can a national government coordinate an effort such as this at the regional and local scale in a way that may appear to be uneven and/or inequitable? How can this be achieved when regional/state/province and local protected areas are often acquired based on availability, willingness of the landowner to sell, cost, and other local factors? The authors note that in Canada, most protected areas are spread out evenly across the political landscape (can you please

provide citations for this?), however this is not the case for many countries. Do the authors have suggestions on how to implement this?

Response 9: This is a really important point. Prior to this study, there was no estimate for the extent to which existing protected areas capture and protect Canadian species/biodiversity, nor did we have any idea of how important it was to coordinate the expansion of protected areas at a national scale in Canada. The extent to which our results can be used to influence how we prioritize and expand protection in Canada is a great question. As you point out, many protected areas are established based on availability, cost, etc., and these factors will undoubtedly remain important moving forward. What our results do, is provide organizations with the ability to add information on how proposed protected areas contribute to the protection of biodiversity at the national scale into the decision-making process. What this does is enable better prioritization at the federal level to protect areas of extremely high biological importance, help identify potential synergies between regions, and enable progress tracking and indicator reporting to identify and address gaps in protection. Our results also help conservationists gauge the extent to which the establishment of novel protected areas measures up against what is possible under a best-case scenario for biodiversity. These ideas are summarized in L176-180 with a modification of past writing. The two new paragraphs at the end before the conclusion also address how our results can be used in the context of other 30x30 priorities (L249-268)

Comment 10:

One concern I have is that the framing makes it seem that national is best when the national and regional approaches are actually prioritizing different things. The national scenario weighs all taxa evenly when looking to prioritize areas across the landscape, so by the nature of how it is set up will try to maximize its efficiency in selecting areas that are representative across all taxa. It is not surprising that they do well in capturing biodiversity. On the other hand, the regional approaches either weigh species by endemism or try to spread out priority areas evenly or to achieve ecosystem representation. In these cases, some species will be prioritized higher (so less balanced across all taxa), or the output be spread out the areas across a more diffuse network to capture many areas but not necessarily the best possible configuration. These are all very different approaches to prioritization that have different principles at the root of them, so it is not surprising they are selecting different areas. I think the authors should be more upfront about what national and regional coordination means, and that they are based in different approaches. National approaches could also focus on endemism or spreading out protected areas across the country, so its more about how the effort is coordinated than who is doing it. I would love to see the text updated to reflect this nuance. I was surprised to find it buried in the methods.

Response 10: We agree that it is no surprise that the national scenario is best when we are reporting outcomes at the same national scale. While this could be a limitation, Target 3 and other goals in the GBF are inherently national in scope, so it makes sense to quantify biodiversity and biodiversity targets, indicators, and outcomes at that scale. You are correct that the national scenario weighs taxa evenly. However, we also want to emphasize that the only difference between the national and sub-national prioritizations is the scale at which the prioritization is run (therefore limiting the species list and grid cells in question) and that they are both based on (weighted) endemism and the same prioritization algorithm. On the other hand, the transnational scenario does use different species weights, and weighs species based on how endemic they are

to Canada, to prioritize Canada's contribution to global biodiversity. As suggested, this point has been clarified in the text in L83-86. We've also improved the clarity of our methods regarding the different prioritization runs (see paragraph L513-521).

Comment 11:

I see in the methods that areas of high human footprint and Indigenous lands were excluded from the analysis. I think you need to say this early on in the introduction/results as I was thinking about this as I read the paper. There is no mention or assessment of how 30x30 could also be used to improve human well-being and nature access, so excluding areas of high human footprint would exclude restoration efforts that could benefit both people and nature. Perhaps some discussion on how this paper is focused on more strict protections under 30x30, rather than other conservation efforts that may count under areas based conservation targets. There are criticisms on area based targets on how they overlook human communities, where we could be finding solutions that co-benefit both. So although avoiding these areas in the analysis helps avoid issues of how protection has led to historical conservation legacies of land displacement, it also avoids the conversation that working in co-production with communities and Indigenous Peoples and Local Communities (IPLC) can be a pathway towards 30x30 as well. In relation to this, the term "wilderness" also has colonial roots and has been considered problematic to some.

Response 11: Great comment. We now mention that we excluded areas of high human footprint and ceded indigenous land in the first line of the results (L77-80).

Additionally, we have better explained our decision to do so in the discussion and have referenced relevant past work that accomplishes some of these things in Canada (L249-268, see our response to Comments 6 and 7 above for more details). We have also removed the term wilderness from our manuscript.

Comment 12:

I also would like more exploration of the limited amount of overlap between the spatial priority scenarios, only 2.8%, with only 34.2% of land prioritized in only some scenarios. This part was glazed over quickly, stating that it was "sensitive to shifting conservation priorities". It would be great to know why there is little consistency in prioritized land across scenarios, and perhaps an understanding of the implications of this.

Response 12: Good point. We have expanded on this finding in L136-139.

Comment 13:

Species Distribution models:

I have a few comments on the underlying species distribution models that require more information

1) It seems that all species were trained on the same geographic extent. If so, this can have serious implications for model outputs, especially for more range restricted species. The spatial extent of the training region should be tailored to the species realistic geographic space. See citations below

(VanDerWal, J., Shoo, L. P., Graham, C., & Williams, S. E. (2009). Selecting pseudo-absence data for presence-only distribution modeling: how far should you stray from what you know?. *Ecological modelling*, 220(4), 589-594.)

(Saupe, E. E., Barve, V., Myers, C. E., Soberón, J., Barve, N., Hensz, C. M., ... & Lira-Noriega, A. (2012). Variation in niche and distribution model performance: the need for a priori assessment of key causal factors. *Ecological Modelling*, 237, 11-22.)

Response 13: We thought a lot about this and many other considerations for building SDMs. The major issue with restricting the geographic extent is that the input data is heavily biased. So much so, that for most range-restricted species, restricting the modelling extent would eliminate areas that are potentially suitable but lack observations due. This is especially true for rural northern regions. To overcome this, we used bias-correction, which was the most influential decision made in the modelling pipeline. We outline our approach to bias correction in L383-420 of the methods.

Comment 14:

2) Did you do any filtering of the occurrence data to account for spatial autocorrelation, bias, and model overfitting?

Response 14: Yes, we filter the data in two ways to reduce autocorrelation and overcome bias. First, we filtered input point data into grid cells (one occurrence per cell) and then further filtered those grid cells when fitting models based on the prevalence of the particular species to reduce spatial autocorrelation, especially of common species. Additionally, we also masked urban areas and water bodies and removed extreme outliers. We improve the explanation of these procedures in the methods in L386-391.

Comment 15:

3) I don't see information on model testing- Did you run any cross-validation or similar testing and if so, did you use any geographic data partitioning for model assessment?

Response 15: We performed a range of model testing but given the extreme input data bias and the way that we used the models in the spatial prioritization, standard x-fold cross-validation with a metric such as AUC (which is already flawed given the lack of true zeros) does not provide a very accurate picture of how realistic the models are. We have another manuscript in the works that deals directly with these issues, but the solutions are too complex and varied to include in this paper, and do not substantially impact the results for this specific application. To summarize the findings briefly, if we do a typical validation on the highly biased input data, we retrieve high AUC as expected in areas that are well-sampled, but models repeatedly predict low suitability in areas in which we know that species are present. For this manuscript, we tried to account for bad models and erroneous predictions in 2 main ways. First, we clipped predicted species distributions by their expertly estimated geographic ranges (i.e., IUCN polygons), as a way of controlling for both species with restricted ranges as well as the limit of model cross-validation. And second, we used ensemble models (averaged BRT and MaxEnts, see following comment and response).

However, in response to your comment and acknowledging even flawed validation is still useful, we have added AUC and PRC scores, for all species, across all of Canada as well as different geographic subsets in the supplementary information in Table S2. Across all species, we were able to predict species observed presences with a high degree of reliability suggesting that our models do a good job at predicting species where they have been observed in Canada. Since it is impossible to quantitatively validate areas without species observations, we also hand checked output distribution maps against existing range maps for all vertebrates, 10% of plants, and as many butterflies as possible. Where necessary we also consulted experts to determine the validity of model predictions to ensure accuracy. This has been added to the methods in L412-420 and L433-443.

Comment 16:

4) what was the reasoning for averaging the BRT and Maxent and in the proportions that you did ? Please provide more information and/or citations.

Response 16: We averaged BRT and Maxent outputs because they both produce sensible present-day projections (the GAM and linear models not included could not account well for the interactions between climate and land cover), but produced slightly different results (again, mainly due to the differences in dealing with biased input data). We trusted the BRTs a bit more in producing sensible northern range extents as the influence of bias-correction was stronger, which is why we split the proportions as we did. We averaged rather than putting both layers into zonation simply to limit the number of input layers and wall time, and ease of final calculations of % species captured in the solution. We acknowledge that this is subjective, but felt it was the best decision given the circumstances and given our results it is highly likely that the priority rank of cells across Canada is extremely stable to these decisions on averaging weights. There is no paper that fits exactly with what we are trying to achieve, but we do cite the papers showing how well BRT models typically work. This has all been clarified in L429-433.

Comment 17:

Minor comments-

Please add the time period for future climate change along with the RCP around line 331 (I note it is mentioned later, but would be good to have included with the data description).

Response 17: This has been added to L365.

Comment 18:

Line 33- there is mention that SDM output discrepancy comes from the SDM rather than the GCP or RCP. However, there is much divergence in climate scenarios across GCMs after the 2050 time period. Given you are using an ensemble and looking at 2080, your model outputs will mute some of these differences. There is support in the literature that using ensemble methods may not be representative of real conditions, and that a range of GCMs should be used to get a sense of prediction uncertainty (see Buisson, L., Thuiller, W., Casajus, N., Lek, S., & Grenouillet, G. (2010). Uncertainty in ensemble forecasting of species distribution. *Global Change Biology*, 16(4), 1145-1157.). At the least the authors should acknowledge this if they are not able to present information on uncertainty of their model outputs.

Response 18: We acknowledge that by using an ensemble GCM we are minimizing some differences between the GCMs for the later time periods. We add this caveat more clearly in the

manuscript (L369-374 and 423-427). We did not include multiple GCMs as we already were dealing with many thousands of runs and really wanted to focus on the decisions that might impact the final priority areas. Given the way that the prioritizations were run (e.g. weighting stable areas more highly), the future projections did not actually change the prioritizations substantially enough to impact the main findings beyond using the present (although certainly would for some species and areas).

Comment 19:

SDMs

Were all the default settings used in Maxent and BRT using the dismo package? please provide more details so that the modeling methods can be fully assessed and/or repeated.

Response 19: This information has been added to lines 408-411 in the methods.

REVIEWERS' COMMENTS

Reviewer #1 (Remarks to the Author):

The authors have addressed the bulk of reviewers' comments. One minor note: regarding the sentences: "Interestingly, our results contrast with past studies identifying important climate refugia in Canada^{32,33}, likely reflecting their focus on climatic stability of the land or predesignated refugia compared to our approach of climate resiliency at the individual species level. In our case, the high priority areas we identify depend on both endemism and climate resilience, thereby retaining complementarity and irreplaceability - core principles of systematic conservation planning.", I would suggest that the authors omit "in our case", since this may imply to many readers that the previously cited works do not consider complementarity and irreplaceability, which is incorrect.

Reviewer #2 (Remarks to the Author):

Thank you for the opportunity to review the revised manuscript of '30x30 biodiversity gains rely on national coordination'. I appreciate the time that the authors took in their responses. For the most part, I feel that the authors have updated the text to reflect the reviewer comments and made clarifications where needed. I can't seem to access the original submission, but when reading this version through there was improved clarity.

I still do find some concerns with the underlying models being trained on such a large geographic extent without accounting for a species range/occurrence within that extent. This could have implications for model outputs, especially for more range restricted species, and can lead to overly simplified outputs and implications for future projections. The authors note that they have undergone several steps to address bias in the data and that they use the larger extent to account for poor sampling in the north of Canada which I agree is an issue. The authors should consider a bias-corrected null model which would better address the bias issue in my opinion (see Raes and ter Steege). However, given the expert review process was undertaken for model outputs and overall the modeling is sound, I will defer to saying that the models are sufficient for their use here. I will suggest that the authors should consider this approach for the other paper they refer to in their letter.

Raes, N., & ter Steege, H. (2007). A null-model for significance testing of presence-only species distribution models. *Ecography*, 30(5), 727-736.

Overall this revised version is much improved and would contribute meaningfully to the area based conservation target literature.

Responses to Reviewer Comments

Comment 1:

Reviewer #1 (Remarks to the Author):

The authors have addressed the bulk of reviewers' comments. One minor note: regarding the sentences: "Interestingly, our results contrast with past studies identifying important climate refugia in Canada^{32,33}, likely reflecting their focus on climatic stability of the land or predesignated refugia compared to our approach of climate resiliency at the individual species level. In our case, the high priority areas we identify depend on both endemism and climate resilience, thereby retaining complementarity and irreplaceability - core principles of systematic conservation planning.", I would suggest that the authors omit "in our case", since this may imply to many readers that the previously cited works do not consider complementarity and irreplaceability, which is incorrect.

Response 1: Thank you for all your comments throughout the review process. In response to your final comment, we have removed "in our case" from the text per your suggestion.

Comment 2:

Reviewer #2 (Remarks to the Author):

Thank you for the opportunity to review the revised manuscript of '30x30 biodiversity gains rely on national coordination'. I appreciate the time that the authors took in their responses. For the most part, I feel that the authors have updated the text to reflect the reviewer comments and made clarifications where needed. I can't seem to access the original submission, but when reading this version through there was improved clarity.

I still do find some concerns with the underlying models being trained on such a large geographic extent without accounting for a species range/occurrence within that extent. This could have implications for model outputs, especially for more range restricted species, and can lead to overly simplified outputs and implications for future projections. The authors note that they have undergone several steps to address bias in the data and that they use the larger extent to account for poor sampling in the north of Canada which I agree is an issue. The authors should consider a bias-corrected null model which would better address the bias issue in my opinion (see Raes and ter Steege). However, given the expert review process was undertaken for model outputs and overall the modeling is sound, I will defer to saying that the models are sufficient for their use here. I will suggest that the authors should consider this approach for the other paper they refer to in their letter.

Raes, N., & ter Steege, H. (2007). A null-model for significance testing of presence-only species distribution models. *Ecography*, 30(5), 727-736.

Overall this revised version is much improved and would contribute meaningfully to the area based conservation target literature.

Response 2: Thank you for your help throughout the review process and for your thoughtful comments and suggestions. Your final point about the geographic extent on which the SDMs were trained is valid and we appreciate your understanding of the challenges we faced and the methods we used to overcome those challenges. Your suggestion of a Null Model approach is a great idea and could help streamline the modelling process in data deficient landscapes. We agree that our extensive review process of model outputs helps ensure valid use of these models for the purpose of this study. We will absolutely consider a null approach for the other paper focusing on bias correction and projection of SDMs.